# Advances in Visualizing Microglial Cells in Human Central Nervous System Tissue

**DOI:** 10.3390/biom12050603

**Published:** 2022-04-19

**Authors:** Christopher E. G. Uff, Karishma Patel, Charming Yeung, Ping K. Yip

**Affiliations:** 1Centre for Neuroscience, Surgery and Trauma, Blizard Institute, Barts and The London School of Medicine and Dentistry, Queen Mary University of London, London E1 2AT, UK; c.uff@qmul.ac.uk (C.E.G.U.); c.yeung@smd21qmul.ac.uk (C.Y.); 2Department of Neurosurgery, Royal London Hospital, Whitechapel, London E1 1FR, UK; 3School of Life Sciences, University of Nottingham, Nottingham NG7 2TQ, UK; mbykp5@nottingham.ac.uk

**Keywords:** microglia, neuroinflammation, immunostaining, immunohistochemistry

## Abstract

Neuroinflammation has recently been identified as a fundamentally important pathological process in most, if not all, CNS diseases. The main contributor to neuroinflammation is the microglia, which constitute the innate immune response system. Accurate identification of microglia and their reactivity state is therefore essential to further our understanding of CNS pathophysiology. Many staining techniques have been used to visualise microglia in rodent and human tissue, and immunostaining is currently the most frequently used. Historically, identification of microglia was predominantly based on morphological structure, however, recently there has been a reliance on selective antigen expression, and microglia-specific markers have been identified providing increased certainty that the cells observed are in fact microglia, rather than the similar yet distinct macrophages. To date, the most microglia-specific markers are P2Y12 and TMEM119. However, other microglia-related markers can also be useful for demonstrating activation state, phagocytic state, and for neuroimaging purposes in longitudinal studies. Overall, it is important to be aware of the microglia-selectivity issues of the various stains and immunomarkers used by researchers to distinguish microglia in CNS tissue to avoid misinterpretation.

## 1. Introduction

Neuroinflammation has recently been identified as a crucially important pathological process in most if not all central nervous system (CNS) diseases. Since the main contributor of neuroinflammation is the microglia that modulate the innate immune response, an understanding of their pathophysiological roles in CNS disease will likely be essential to the discovery of effective therapies [1]. However, it has previously been difficult to clearly distinguish microglia from meningeal, perivascular, choroid plexus, and circulating macrophages in pathological conditions [2].

The origin of microglia was for a long time though to derive from blood monocytes in the bone marrow, however in 1999 Alliot et al. showed that microglia were in fact derived from progenitors originating in the yolk sac [3]. This was further confirmed using fate-mapping analysis that showed that microglia migrate from the embryonic yolk sac at approximately E9.5 in mice and 4.5 weeks gestation in humans [4]. After the microglia replicate, they spread throughout the whole CNS in a cellular grid formation until adulthood, eventually comprising approximately 10% of the total glial cell population [5,6].

Currently, there is a growing body of literature that recognises the importance of microglia and their functional roles in CNS homeostasis and pathophysiology. During homeostasis in the adult, microglia play several key roles: the long and thin branching processes provide constant surveillance of the local environment to maintain tissue integrity, such as repairing the blood vessels [7,8], and permitting synaptic plasticity by engulfing synapses [9]. Furthermore, connections between perivascular microglia and endothelial cells, along with pericytes and astrocytes, form the neurovascular unit that maintains blood-brain barrier integrity [10].

Microglia have been implicated as a crucial player in disease pathophysiology. In their normal state, the ramified (‘branch like’)/resting (non-reactive phenotype) microglia exhibit a morphology of very thin and long processes with a small cell body (Figure 1A–C, Figure 2A–C, Figure 3A–C, Figure 4A–C, Figure 5A–C and Figure 6A–C, unpublished data). In pathological conditions, the microglia undergo a transition that retracts their processes resulting in thicker and shorter processes and a larger cell body, until finally the processes are significantly reduced, which is termed amoeboid (‘round like’)/reactive microglia (Figure 1D–O, Figure 2D–O, Figure 3D–O, Figure 4D–O, Figure 5D–O and Figure 6D–O, unpublished data). Interestingly, in the examples of microglial activation states provided in this review (all seen within a 1.5 mm cross-section of a brain biopsy from a patient suffering severe traumatic brain injury (TBI)), it is possible to see both ramified and various other reactive states of microglia. This suggests that the microglial response is at a cellular level rather than a global cellular response to TBI.

Microglia are thought to be significant contributors to various CNS diseases by releasing cytokines and by displaying phagocytic properties. In amyotrophic lateral sclerosis, there is evidence to support both neuroprotective roles and also neurotoxicity [11]. Impaired microglial activity and altered microglial response to amyloid beta protein are associated with an increased risk of Alzheimer’s disease. However, there is also abundant evidence that activated microglia can be harmful to neurones [12]. In TBI, dysregulation of microglia can release high levels of pro-inflammatory and cytotoxic mediators that contribute to neuronal dysfunction and cell death. However, microglia in the injured brain can also produce neuroprotective factors, clear cellular debris, and orchestrate neurorestorative processes that are beneficial for neurological recovery after TBI [13]. Interestingly, in spinal cord injury, recent evidence suggests that microglia are essential to the formation of the neuroprotective glial scar [14], and that microglial depletion exacerbates injury and impairs functional recovery [15]. Although microglia are deleterious in ischemic stroke, there is also evidence to suggest that microglial activation is critical for promoting functional recovery through neurogenesis, angiogenesis, and synaptic remodeling [16]. Overall, the roles of microglia in CNS injuries are extremely complex and require further investigation.

Histological staining is one of the most common techniques for the identification of any cell type, including microglia, as opposed to Western blotting or enzyme-linked immunosorbent assay (ELISA). Immunohistochemistry staining is a fairly simple procedure to perform, identifies the exact cellular location within the CNS, provides a morphological representation of cellular processes, and simultaneously expresses the intensity level of one to several proteins of interest. Furthermore, it is highly sensitive and selective when a particular immunostaining method is used. In contrast, Western blotting or ELISA (for example) generates a single signal proportional to the amount of protein present in a homogenised tissue sample.

To understand the development in microglia visualization, this review will explore the historical timeline of microglial staining from 1899 to 2021. The development of markers to identify specific microglia and distinguish them from other cell types will also be discussed. This will be supported by microglia immunostaining using fresh brain biopsy tissue from the superior frontal gyrus of patients suffering severe TBI ranging from 1h to 146 h post injury. Fresh brain biopsies were obtained prior to insertion of an intracranial pressure transducer using a brain biopsy cannula as previously described [17]. Contused tissue was obtained from a patient suffering severe TBI who underwent therapeutic resection of contused brain tissue as part of life-saving treatment. All fresh brain tissue was quickly transferred into formalin the next day and was submerged in 20% sucrose. To study peripheral macrophages, clotted blood (acute subdural haematoma) from a neurosurgical (brain) operative field was collected and processed in the same way as the fresh brain tissue. Thereafter, the biopsies and the blood clots were simultaneously cut cross-sectionally using a cryostat at 12 μm and mounted onto microscope slides. From this, we determined the most successful stains for imaging microglia in humans and rodents. We also highlight some of the issues to consider in order to avoid misinterpretation. 

## 2. History of Microglial Staining

Research in this field was pioneered by Franz Nissl, Santiago Ramón y Cajal, and Pío Del Río Hortega, with the latter scientist rewarded with the ‘father of microglia’ accolade [18]. During the time of the first microglia discovery, different methods of impregnation of brain tissue with various stains to identify cells in nervous tissue were used [19]. In 1899, Franz Nissl, using his eponymous Nissl stain, was the first to identify cells he termed stäbchenzellen that were rod-like microglial cells, but he did not pursue this further [20]. It was not until 1913 that the origins of microglia were revealed when Ramón Y Cajal used a silver chromate staining technique to identify a group of cells that were non-neuronal and non-astrocytic in morphology [21]. However, it was Pío Del Río Hortega who in 1919, used a modified silver carbonate impregnation method that stained a selective group of glial cells, and named one of these new populations of cells microglia [8,22]. He also introduced the concept of plasticity in microglia after discovering that they could change their morphological shape depending on the brain’s pathological state.

Identification of microglia is facilitated by the use of appropriate staining techniques, and the suitability of a stain is based on its ability to discriminate microglia from other CNS cell populations including neurons, astrocytes, endothelial cells, and macrophages. Furthermore, stains must have sufficient ability to stain either all microglia, or only certain activation states [8]. Del Río-Hortega’s ammoniacal silver carbonate method not only stains microglia but also oligodendrocytes, and it is a complex protocol and requires optimum conditions for success [23]. Therefore a more selective, quicker, and easier staining protocol is required. Furthermore, relying on cell morphology can be problematic since microglia transform into different morphological shapes depending on their activation state. 

## 3. Microglia Staining Development 

Albert Coons and colleagues were the first to demonstrate selective staining through antigen-antibody interactions using an antibody and fluorescent dye [24]. Thereafter, other immunohistochemical techniques using enzyme labels such as peroxidase and alkaline phosphatase were developed [25]. This revolutionised the histological field by offering a more detailed spatiotemporal expression of specific antigens, apart from solely relying on morphology. Since the mid-1900s, researchers have tried many various markers ranging from ATPase to the recent TMEM119 to selectively identify microglia (Table 1).

### 3.1. ATPase

Chronologically, the Adenosine triphosphatase (ATPase) stain was first used in the identification of microglial cells beyond Hortega’s silver chloride method. The enzymatic marker ATPase was discovered in the 1970s to be among the most selective enzymatic markers for microglia in rat and rabbit brain tissues, obtained from the normal brain and after a stab wound injury [26]. Although the cytochemical staining protocol using the Wachstein-Meisel medium was successfully used to identify microglia, ATPase was also located in various other cellular regions such as vessel walls, astrocytes, oligodendrocytes, and proximal axons [26]. This is not surprising as ATPase is an essential enzyme for energy, active transport, and pH homeostasis in all cell types [27,28]. Furthermore, other studies that used this method demonstrated the presence of ATPase in rat placenta, kidney, heart, prostate, uterus, and leg muscle [29]. Overall, these findings suggested that ATPase staining cannot be used as a specific marker for microglia.

### 3.2. Iba1

The Ionized calcium-binding adaptor molecule 1 (Iba1) immunostain was a significant breakthrough in terms of imaging and identification of microglial cells. Iba1 is located in a major histocompatibility complex (MHC) class III region, encoding an EF-hand protein [30]. A research study in 1998 suggested this novel calcium-binding protein Iba-1 was a microglia-specific marker [31]. Moreover, Iba1+ immunostaining can clearly label the cell body and the fine processes in ramified microglia, and that reactive microglia express greater Iba1+ levels compared to non-reactive ramified microglia, and it was therefore less dependent on the morphological shape of ramified or amoeboid [31] (Figure 1, Figure 2, Figure 3, Figure 4, Figure 5 and Figure 6, unpublished data). Since this discovery, the majority of research papers with histological analysis of microglia included the use of Iba1 immunostaining to observe Iba1-positive cells [32]. Although Ito and colleagues suggested the Iba1 marker was microglia-specific, research has suggested that Iba1 is also an optimum marker for macrophages in both rats and humans [33,34]. Since reactive microglia and macrophages exhibit similar phenotypes, it is reasonable to assume that they express similar molecules. Overall, Iba1 can selectively distinguish from other glial cells within the CNS, but it is unable to distinguish between microglia and macrophages as both have high Iba1 expression, so care is required when interpreting Iba1+ immunostained cells in human post-mortem and/or animal tissue. However, most highly reactive microglia will exhibit some form of extrusion in comparison to the spherical amoeboid-shaped macrophages (Figure 1M−R, Figure 2M−R, Figure 3M−R, Figure 4M−R, Figure 5M−R and Figure 6M−R, unpublished data) [35]. Therefore, using morphological differences (i.e., microglia exhibit processes and macrophages do not exhibit processes) in combination with location within the tissue (i.e., microglia are within the intact CNS tissue and macrophages are on the periphery or within vasculature) affords the possibility of using Iba1 to stain for microglia within the CNS.

### 3.3. HLA-DR

Another MHC-related microglia marker is the Human Leukocyte Antigen—DR isotype (HLA-DR). This glycoprotein complex is an antigen-presenting molecule that was initially found as an MHC class II cell surface receptor on antigen-presenting cells such as macrophages, dendritic cells, and B-lymphocytes, so immunostaining is predominantly outer membrane labeling [36]. It was present on activated microglia in human Parkinson’s disease post mortem brain tissue, based on the morphology originally described by Del Rio Hortega and Penfield [37], however, McGeer and colleagues were careful to mention the HLA-DR+ staining observed in the reactive microglia, and also in brain macrophages, in the discussion. Interestingly, Gehrmann and colleagues demonstrated HLA-DR was strongly present in resting/ramified microglia, especially in the white matter [38], however, careful observation of the images of putative ramified microglia, immunopositive for HLA-DR, reveals slightly reactive microglia based on the shorter and thicker processes, in comparison to the long and thin processes observed in true ramified microglia found in normal conditions. In this review, co-immunostaining with Iba1 (Wako, Alpha Laboratories Ltd, Cat. No. 019-19741) and HLA-DR (Invitrogen, ThermoFisher, Horsham, UK, Cat. No. MA5-11966) showed low HLA-DR+ immunostaining in ramified microglia (Figure 2A–C), but increased immunostaining in reactive microglia (Figure 2D–O) and in macrophages (Figure 2P–R) (unpublished data). Overall, HLA-DR can identify reactive microglia if the microglia exhibit processes, however, it is difficult to distinguish them from other peripheral immune cells such as monocytes-derived macrophages when in the full reactive state.

### 3.4. CD68 

A commonly used marker in microglia research is the Cluster of Differentiation 68 (CD68). This molecule is an endosomal/lysosomal transmembrane glycoprotein, highly expressed in macrophages and other mononuclear phagocytes, so immunostaining is predominantly lysosomal labeling within the cytoplasm [39]. Interestingly, it has been used frequently to study microglia in human tissue with neurodegenerative disease or head injury [32,40,41,42]. CD68 has been shown to label activated phagocytic microglia as it is upregulated in active phagocytic cells in human post mortem brain tissue [41]. Similarly, in amoeboid microglia, there was increased expression of CD68 [32]. In this review, co-immunostaining with Iba1 (Wako, Alpha Laboratories Ltd, Cat. No. 019-19741) and CD68 (Invitrogen, ThermoFisher, Horsham, UK, Cat. No. 14-0688-82) showed CD68+ immunostaining in ramified microglia (Figure 2A–C), and in reactive microglia (Figure 2D–O) and macrophages (Figure 2P–R) (unpublished data), and this agrees with published CD68+ microglial staining [32]. However, there are also published reports that ramified microglia do not express CD68 [42]. Therefore, it remains to be resolved whether CD68 is present in ramified microglia. One important point is that the circulating macrophages have a much stronger CD68+ expression than microglia. Similar to other markers mentioned, CD68 expression is also present in other cell types including macrophages, neutrophils, fibroblasts, and activated endothelial cells, so is not specific to microglia [43,44]. Another issue with CD68 is that it can not be used to identify microglial processes due to its low expression in these regions and high expression at the lysosomal membrane in ramified and amoeboid microglia near the nucleus [32]. For this reason, relying solely on the CD68 expression may have influenced the interpretation of microglial involvement and reduced the validity of these markers as effective microglia-specific markers. It would however make a useful pan-macrophage or phagocytic marker. 

### 3.5. Galectin-3 

Galectin-3 is described as a carbohydrate-binding protein glycoprotein located in different cell and tissue types [45]. In normal conditions, the galectin-3 level in ramified microglia is low, but activated microglia have upregulated galectin-3 expression both intracellularly and outside the cell, which suggests that it plays a key role in the microglia’s immune response following TBI in rodents [46]. Furthermore, in this review, co-immunostaining with Iba1 (Wako, Alpha Laboratories Ltd, Cat. No. 019-19741) and galectin-3 (R & D Systems, Bio-Techne Ltd, Abingdon, UK, Cat. No. AF1197) showed low, if any galectin-3+ immunostaining in ramified microglia (Figure 3A–C), but it was increased in reactive microglia (Figure 3D–O) and macrophages (Figure 3P–R) (unpublished data). Since the immunostaining is predominantly in the cytoplasm, it is unable to label microglial processes. Interestingly, galectin-3 has been shown to control microglia morphology and phagocytosis [47]. However, there is evidence for the expression of galectin-3 in the nucleus and cytoplasm of macrophages [48], neutrophils, T-cells, and mast cells [49]. Therefore, using galectin-3 as a sole marker for microglia is limited due to low levels in normal conditions and lack of specificity, but can be used to determine the development of reactive microglia.

### 3.6. P2Y12

The adenosine diphosphate (ADP) purinergic receptor P2Y, G protein-coupled 12 (P2Y12/P2RY12) is a chemoreceptor for ADP, which is G-protein coupled and was found to be selectively expressed by microglia and absent in monocytes or macrophages [50,51,52]. This recently identified marker is selective for microglia: the anti-P2Y12 antibody (Sigma-Aldrich, Cat. No. HPA014518) has been used to selectively label parenchymal microglia in human CNS tissue [53] and the anti-P2Y12 antibody (AnaSpec, Cambridge Bioscience, Watford, UK, Cat. No. ANA55043A) has been used to label microglia in the rodent spinal cord [54] and the human brain (Figure 5A–O, unpublished data). Interestingly, a study by Mildner and colleagues of paraffinised post mortem brain tissue using these techniques found that psychiatric diseases such as schizophrenia did not affect levels, but neuroinflammatory diseases such as multiple sclerosis and Alzheimer’s disease showed a decrease in P2Y12 expression [53]. Furthermore, upon immunofluorescent staining of MS lesions, there was no expression of P2Y12 within the lesion, however, at the lesion border the P2Y12 cells were visible and had activated morphology [53]. Another finding from Zrzavy and colleagues supports this concern that P2Y12 staining was reduced in microglia in MS patients indicating an inability of the stain to detect microglia in their active state [55]. Furthermore, P2Y12 was strongly expressed in ramified microglia in their homoeostatic state, but was reduced in reactive microglia [50]. Interestingly, in this review, strong P2Y12+ staining was observed in all microglia activation states with labeling in both cytoplasm and processes (Figure 4A–O) and was not present in macrophages (Figure 5P–R) (unpublished data). Overall, P2Y12 seems to be a true microglia marker that can successfully identify ramified microglia, however there are concerns regarding its ability to identify reactive microglia. However, based on unpublished data from acute TBI in human brain tissue (Figure 4), P2Y12 can identify reactive microglia, suggesting this marker may be potentially more suited to acute rather than chronic neurological diseases.

### 3.7. TMEM119

Recently, several studies have identified putative microglia-specific candidates using flow cytometry and deep RNA sequencing or microarray analysis [56,57]. However, it was Bennett and colleagues who were able to systematically validate the microglia-specific candidates and identify transmembrane Protein 119 (TMEM119) as the most valid and stable microglia-specific marker in humans and rodents [58]. In the search for a new tool to study microglia, Bennett and colleagues developed commercially available antibodies that were selective for TMEM119 (Abcam, Cambridge, UK, Cat. No. ab209064 or ab210405 for mouse, and ab185333 for human), which detected most, if not all, microglia, although it did not detect macrophages or other immune or neural cells [58]. Bennett and colleagues used TMEM119-GFP mice to specifically label TMEM119+ microglia and created conditional knockout TMEM119-Cre mice to inactivate the *TMEM119* gene to demonstrate that murine microglia mature by postnatal day 14 [58]. Furthermore, the generation of the TMEM119-GFP mice provides a much more convenient and reliable identification of microglia for studying since other mouse lines using promoters such as LysM, CD11b or Cx3cr1 do not exhibit microglia-specific labeling [59,60,61]. In the same year, the view that TMEM119 was microglia-specific was independently supported by Satoh and colleagues, who demonstrated TMEM119 (Sigma-Aldrich, Merck Life Science UK limited, Gillingham, UK, Cat. No. HPA051870) as the most promising candidate for human microglia-specific markers [62]. Interestingly, the TMEM119 marker expression was not elevated by exposure to lipopolysaccharide, IFNγ, IL-4, IL-13, or TGFβ1 in vivo [62]. Overall, TMEM119 is currently one of the most promising microglia-specific markers. In this review, similar to P2Y12 immunostaining, strong TMEM119 was observed in ramified and all microglia activation states as it labels both the cytoplasm and processes (Figure 5A–O) and is not present in macrophages (Figure 5P–R) (unpublished data). However, to date, there is no suggested role for TMEM119 located on the cell surface, and the effect it may have on microglia function, such as during ramified or activated states, have yet to be fully determined. 

### 3.8. CD11b (Clone OX42)

Cluster of Differentiation 11b (CD11b), also known as integrin subunit alpha M (ITGAM) and complement receptor 3 alpha (CR3A) is an alpha subunit of the integrin complement receptor part 3, which is involved in adhesion processes and the uptake of complement-coated molecules [63]. Some antibodies that can detect CD11b is the clone OX42, thus the term OX42 is often used rather than CD11b in the detection of elevated microglial CD11b in CNS tissue [64,65,66,67,68]. OX42+ immunostaining is predominantly located in the cell body and microglial processes [51,52,56,57]. In contrast, the anti-rat CD11b staining observed in this review was more restricted to the cell bodies and was limited in the ramified state (Figure 6A–C), although with some expression in more activated states (Figure 6D–I) (unpublished data). However, as the target species for anti-rat CD11b (1:100, Bio-Rad Laboratories Ltd, Watford, UK, Cat. No. MCA275R) is the rat, potentially the immunostaining observed was non-specific staining. More concerning is the anti-human CD11b (1:400, Novus Biologicals, Bio-Techne Ltd, Abingdon, UK, Cat. No. MAB16991), which according to the company website and datasheet is a microglia marker (https://www.novusbio.com/products/cd11b-antibody-238446_mab16991, accessed on 1 April 2022). In this review, the anti-human CD11b staining observed was present in the cell body and processes of some cells in the human brain tissue (Figure 6J–R). However, there was no coexpression of anti-human CD11b when co-immunostained with the microglia markers Iba1 and P2Y12 (Figure 6J–O), although anti-human CD11b was coexpressed in cells stained with the astrocyte marker GFAP (Figure 6P–R). Overall, there seem to be no reliable CD11b antibodies available to immunostain microglia in human tissue, and care is required when using an antibody that has not been fully characterised as it may provide non-specific binding or binding onto non-microglial cells.
biomolecules-12-00603-t001_Table 1Table 1Microglia immunostaining markers. A selected list of antibodies used to study microglia in immunohistochemistry. NOTE: *, requirement to use high Triton X-100 concentration (0.5%) for this specific antibody; **, no routine gross macroscopic abnormalities. Abbreviation: AD, Alzheimer’s disease; MS, Multiple sclerosis; TBI, Traumatic brain injury; SCI, Spinal cord injury.MarkerAntibody UsedCompany and Cat. No.Concentration UsedSpecies DiseaseReferencesIba1Polyclonal rabbit anti-Iba1Wako, 019-197411:500Rodents & Human brainAD; MS; SCI; TBI; Sepsis [32,54]; In current reviewPolyclonal goat anti-AIF1/Iba1Novus Biologicals, NB100-10281:250Human brain TBIIn current reviewHLA-DRMonoclonal mouse anti-HLA-DR, DQ, DP (clone CR3/43)DAKO, M07751:100Human brainNormal **; MS[38,55]Monoclonal mouse anti-HLA-DR (clone LN3)Invitrogen, MA5-119661:500Human brainTBIIn current reviewCD68Monoclonal mouse anti-CD68 (clone KP1)DAKO, M08141:400-1:500Human BrainAD; MS [32,41,55]Invitrogen, 14-0688-821:500Human brainTBIIn current reviewGalectin-3Polyclonal goat anti-galectin-3R & D SystemsAF11971:250Rodent & Human brainTBI[46]; In current reviewP2Y12Polyclonal rabbit anti-mouse P2Y12AnaSpec, ANA55043A1:200Rodents & Human brainSCI, TBI[54]; In current reviewPolyclonal rabbit anti-human P2Y12Alomone labs, APR-0121:50-1:200Human brainNormal **; AD[38];Polyclonal rabbit α-human P2Y12Sigma Aldrich, HPA0145181:150Human brainNormal **[53]TMEM119Monoclonal rabbit anti-TMEM119 Abcam, ab209064 *, ab210405, ab1853331:100Mice & Human brainAD, TBI[58]; In current reviewPolyclonal rabbit anti-TMEM119Sigma Aldrich, HPA0518701:100Human brainStroke & MS[55,62]

## 4. Concluding Remarks

This review focused on visualising microglia in CNS tissue samples from rodents to human post mortem or brain biopsies spanning over a century, from the time the term microglia was first coined by Pío Del Río Hortega to the discovery of TMEM119 as a microglia-specific marker. Throughout this period, the morphology of microglia was the main feature that enabled the identification with a small cell body and long thin processes in the resting or homeostatic state. However, when microglia became reactive due to a pathological stimulus, they exhibit morphological change, which in its most activated state becomes amoeboid and can therefore be difficult to distinguish from spherical macrophages. This can be problematic as it is difficult to distinguish whether the neuroinflammation observed is directly from microglia per se, or the effect of circulating macrophages. Therefore, there is a need to find a solution.

Although there have been many microglia markers identified, with some discussed in this review, many were not selective as they also stain macrophages or other cell types. To date, there are only two immunomarkers, namely P2Y12 and TMEM119 that can specifically identify microglia, not only from morphology but also the expression of microglia-specific proteins that are distinct from other cell types. Apart from identifying microglia, recent studies suggest there are now immunomarkers to distinguish if the microglia have become activated and the specific phenotype of microglial reactivity. Although many researchers have used the M1 and M2 phenotype terms to distinguish classical pro- and anti-inflammatory microglia, respectively, this linear concept of microglia polarization is much outdated [69]. This is supported by recent studies showing microglial activation states exist as a 3D stellate polyhedron, which implies that there are multidimensional forms of microglial activation states [70,71,72]. This suggests that different stimuli can induce different microglial responses, so having the ability to distinguish reactive microglia further into the various multidimensional forms would further our understanding of neuroinflammation in various diseases. 

One particular activation state that was recently identified is the disease-associated microglia (DAM). These specifically activated microglia express an increase in the *Iba1*, *Cst3,* and *Hexb* microglia marker genes and a decrease in the *p2yr12*, *Cx3cr1, CD33*, and *Tmem119* homeostatic microglia marker genes [73]. However, these DAM were identified from transgenic mice and post mortem brains with Alzheimer’s disease. This therefore may apply only to chronic microglia activation as Alzheimer’s disease is a chronic neurodegenerative disease, so in comparison to an acute activation response such as TBI, there may be differences observed in the activation status. Furthermore, post mortem tissue may take up to 2 days after death before tissue fixation [74] and inadequate perfusion-fixation of a large tissue mass may explain the existence of certain false-positive pathological observations such as ‘dark’ neurones in post mortem tissue [75,76]. Therefore, these differences may explain the discrepancies between the reduced P2Y12 immunostaining previously reported and P2Y12+ immunostaining from brain biopsies in acute TBI reported in this review. 

We suspect that the neuroinflammation in neurodegeneration and acute traumatic injuries may look similar to currently known staining methods, although could potentially be extremely different. To date, researchers do not have the relevant tools and/or focus in detail to classify neuroinflammation into more definitive subgroups. One solution is to carry out simultaneous multiple immunostainings with various microglia markers to not only confirm cells studied are microglia, but the type of activation states. For example, the use of TMEM119 as a pan-activation state microglia marker can identify cell body and processes, followed by galectin-3 for reactive microglia, CD68 as a phagocytic marker, CD16/32 for classical pro-inflammatory reactive microglia, and arginase-1 for classical anti-inflammatory reactive microglia.

Given that various non-selective microglia markers have been used in research studies, it is imperative to take care in interpreting past research studies as researchers may provide misinterpretation of the data with the assumption the markers were microglia-specific, based on their current knowledge at the time of writing. However, even with the microglia-specific markers available to researchers to date, there are issues to consider. For example, there are thousands of gene expression differences from human microglia of a middle-aged (mean age 53 years old) compared to an aged brain (mean age 94 years old) suggesting that the age of microglia is a variable factor [77]. Furthermore, it was shown that human and mouse microglia age differently based on RNA sequencing data, suggesting care when extrapolating mouse microglia data for human interpretation [78]. Interestingly, the gender of the microglia can also affect its gene expression levels and function. A recent study by Villa and colleagues has shown transcriptomic differences in mice between adult male and female microglia, with female microglia expressing a neuroprotective role in acute focal cerebral ischemia [79]. Overall, research data suggests there are many factors to consider when interpreting microglia data.

## 5. Conclusions

Histologically, morphology has been a strong feature to identify microglia, especially at the ramified/resting/homeostatic state. However, in the amoeboid/reactive state, using markers such as CD68 and HLA-DR can label microglia but also macrophages, thus causing misinterpretation. Therefore, it is imperative to accept the use of additional immunomarkers to definitively determine the microglial cell. With microglia-specific markers available to researchers, we can start to explore the true roles of microglia in homeostasis and neuropathological diseases in acute and/or chronic states. Solving this conundrum would provide a greater understanding of CNS disease and increase the potential for the development of effective therapies. 

## Figures and Tables

**Figure 1 biomolecules-12-00603-f001:**
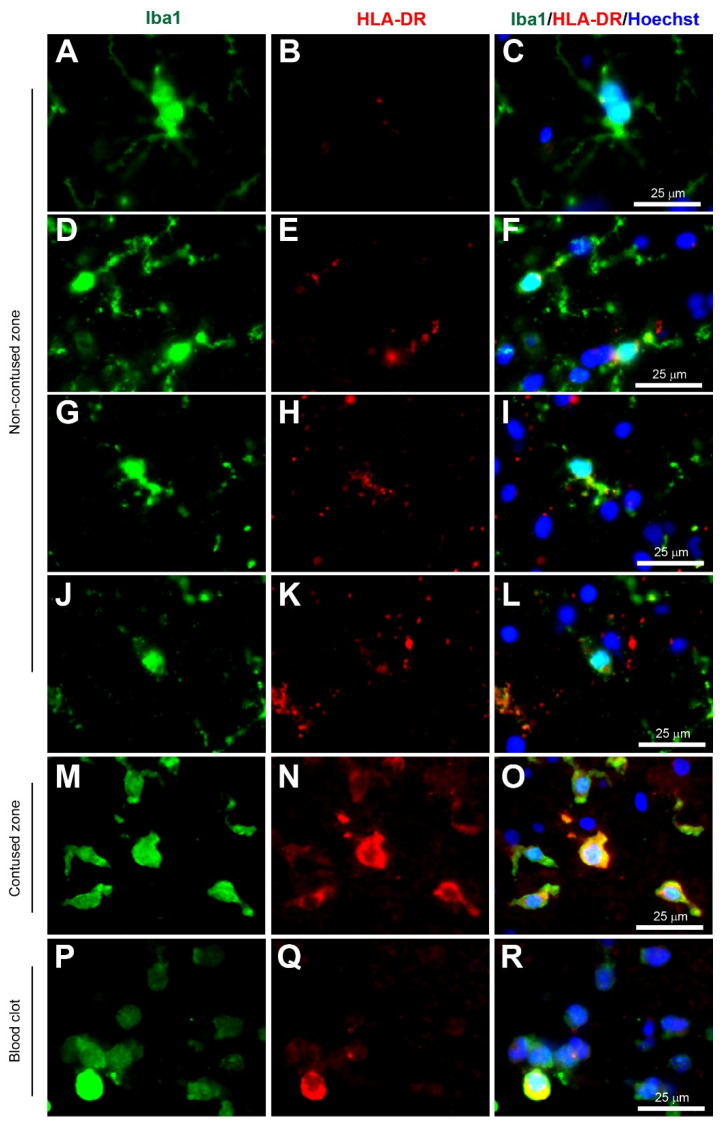
Coexpression of Iba1 and HLA-DR. Human brain biopsy samples from the superior frontal gyrus after TBI were stained with the primary antibodies: rabbit anti-Iba1 (1:1000, Wako, Cat No. 019-19741) and mouse anti-HLA-DR (1:500, Invitrogen, Cat No. MA5-11966) followed by rabbit anti-Alexa Fluor 488 and mouse anti-Alexa Fluor 594, respectively. Limited HLA-DR expression was observed in Iba1+ microglia with a ramified morphology (**A**–**C**), but strong HLA-DR expression in Iba1+ microglia in the activated state (**D**–**L**) in the non-contused brain tissue. Predominantly amoeboid microglia stained with Iba1 and HLA-DR were observed in contused brain tissue (**M**–**O**). Clotted blood from a neurosurgical (brain) operative field was used to identify a few peripherally-derived macrophages that strongly stained for both Iba1 and HLA-DR (**P**–**R**). Scale bars are 25 μm. Unpublished data.

**Figure 2 biomolecules-12-00603-f002:**
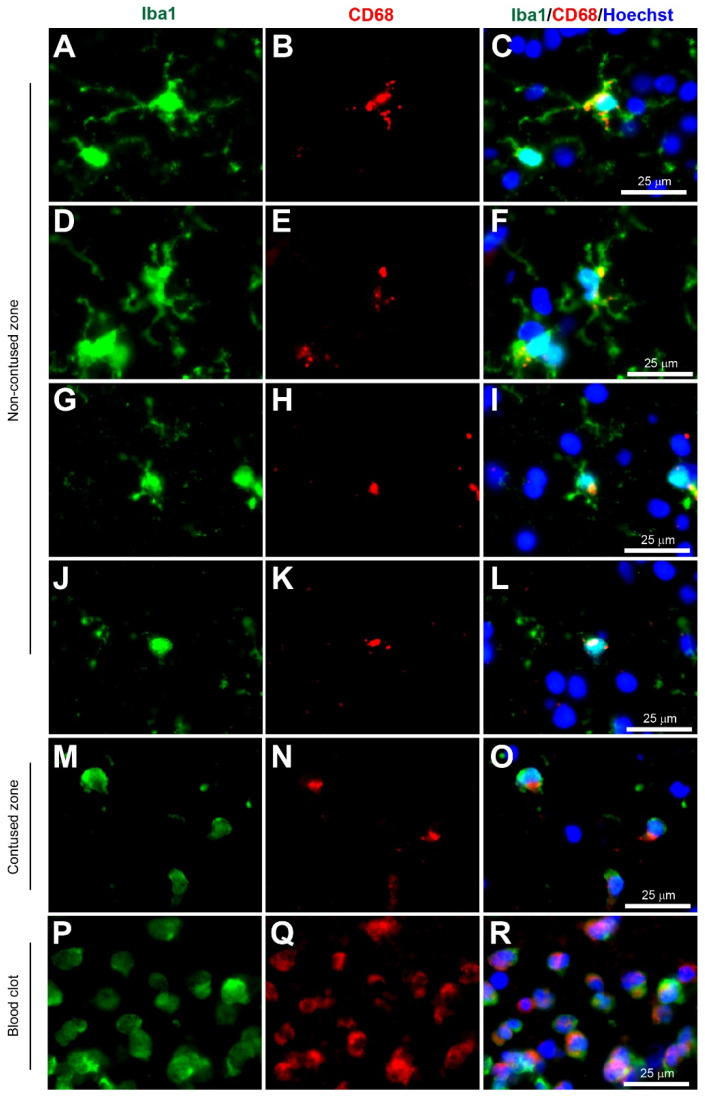
Coexpression of Iba1 and CD68. Human brain biopsy samples from the superior frontal gyrus after TBI were stained with the primary antibodies: rabbit anti-Iba1 (1:1000, Wako, Cat No. 019-19741) and mouse anti-CD68 (1:500, Invitrogen, Cat No. 14-0688-82) followed by rabbit anti-Alexa Fluor 488 and mouse anti-Alexa Fluor 594, respectively. Iba1+ microglia co-expressed with CD68 can be observed in various activated states, ranging from the most ramified state (**A**–**C**) through increasing states of activation (**D**–**F** and **G**–**I**) to the most amoeboid morphology (**J**–**L**) in the non-contused brain tissue. In contrast, predominantly amoeboid microglia stained with Iba1 and CD68 were observed in contused brain tissue (**M**–**O**). Clotted blood from a neurosurgical (brain) operative field was used to identify peripherally-derived macrophages that strongly stained for both Iba1 and CD68 (**P**–**R**). Scale bars are 25 μm. Unpublished data.

**Figure 3 biomolecules-12-00603-f003:**
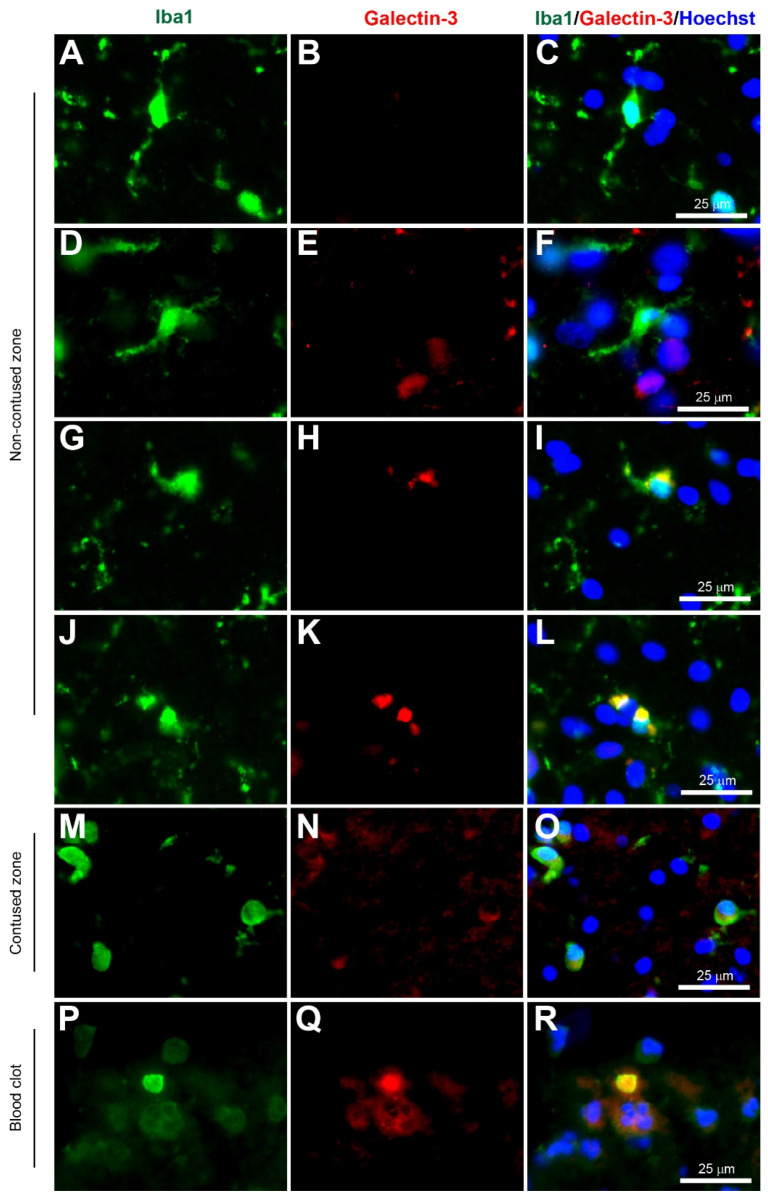
Coexpression of Iba1 and galectin-3. Human brain biopsy samples from the superior frontal gyrus after TBI were stained with the primary antibodies: rabbit anti-Iba1 (1:1000, Wako, Cat No. 019-19741) and goat anti-galectin-3 (1:500, BioLegend, Cat No. 126701) followed by rabbit anti-Alexa Fluor 488 and goat anti-Alexa Fluor 594, respectively. Limited galectin-3 expression was observed in Iba1+ microglia with a ramified morphology (**A**–**C**), but there was a strong galectin-3 expression in Iba1+ microglia in the activated state (**D**–**L**) in the non-contused brain tissue. Predominantly amoeboid microglia stained with Iba1 and galectin-3 were observed in contused brain tissue (**M**–**O**). Clotted blood from a neurosurgical (brain) operative field was used to identify peripherally-derived macrophages that strongly stained for both Iba1 and galectin-3 (**P**–**R**). Scale bars are 25 μm. Unpublished data.

**Figure 4 biomolecules-12-00603-f004:**
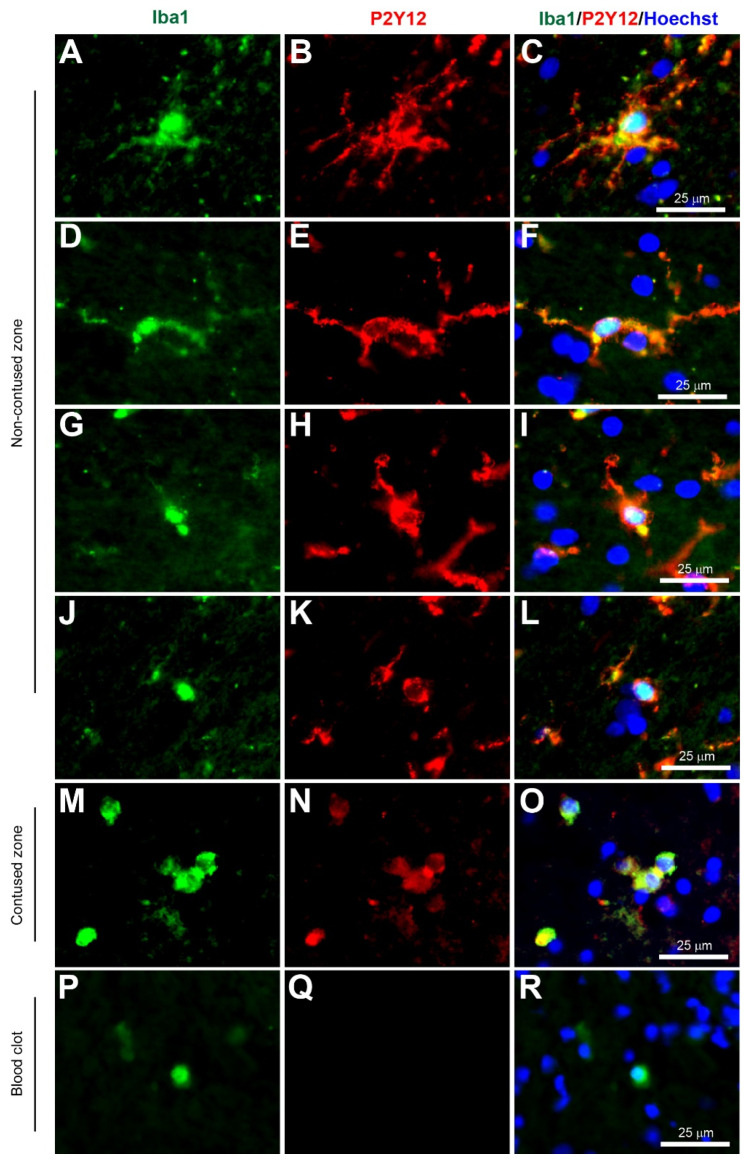
Coexpression of Iba1 and P2Y12. Human brain biopsy samples from the superior frontal gyrus after TBI were stained with the primary antibodies: goat anti-Iba1 (1:500, Novus Biologicals, Cat No. NB100-1028) and rabbit anti-P2Y12 (1:200, AnaSpec, ANA55043A) followed by goat anti-Alexa Fluor 488 and rabbit anti-Alexa Fluor 594, respectively. Strong coexpression was observed in Iba1+ microglia with P2Y12 in both ramified (**A**–**C**) and activated state (**D**–**L**) in the non-contused brain tissue. Predominantly amoeboid microglia stained with Iba1 and P2Y12 were observed in contused brain tissue (**M**–**O**). Clotted blood from a neurosurgical (brain) operative field was used to identify peripherally-derived macrophages that showed strong Iba1+ but lacked P2Y12+ immunostaining (**P**–**R**). Scale bars are 25 μm. Unpublished data.

**Figure 5 biomolecules-12-00603-f005:**
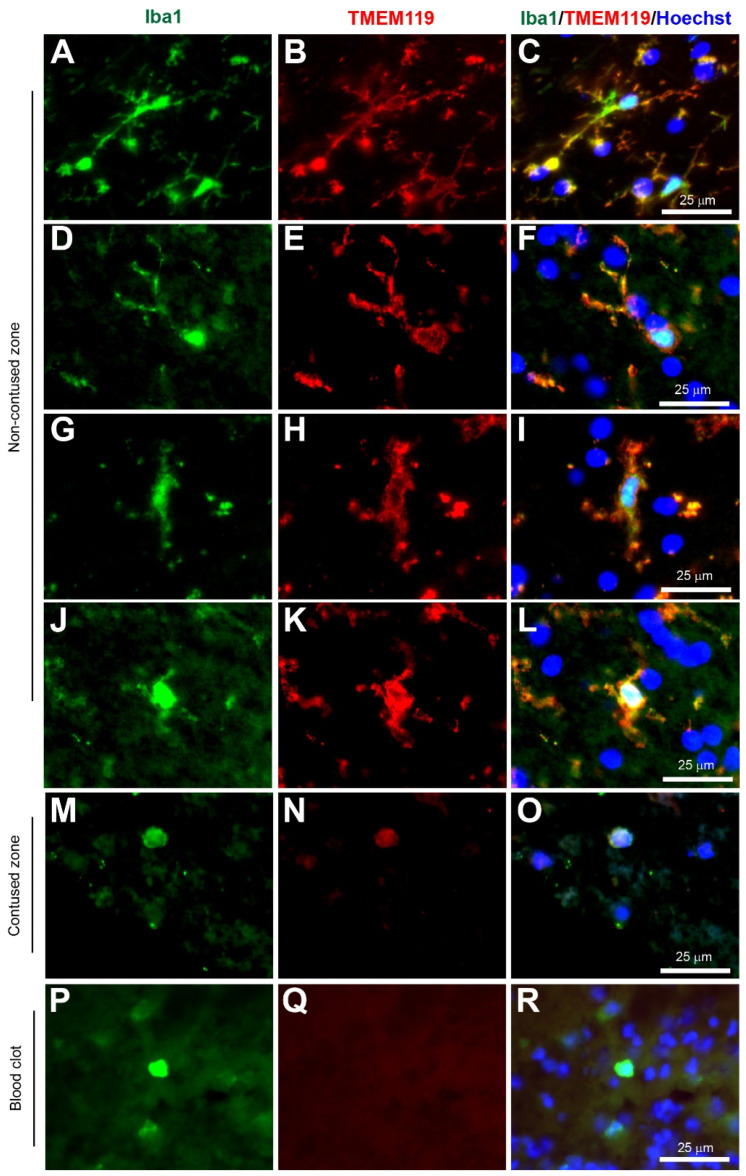
Coexpression of Iba1 and TMEM119. Human brain biopsy samples from the superior frontal gyrus after TBI were stained with the primary antibodies: goat anti-Iba1 (1:500, Novus Biologicals, Cat No. NB100-1028) and rabbit anti-TMEM119 (1:500, Abcam, Cat No. ab185333) followed by goat anti-Alexa Fluor 488 and rabbit anti-Alexa Fluor 594, respectively. Strong coexpression was observed in Iba1+ microglia with TMEM119 in both ramified (**A**–**C**) and activated state (**D**–**L**) in the non-contused brain tissue. Predominantly amoeboid microglia stained with Iba1 and TMEM119 were observed in contused brain tissue (**M**–**O**). Clotted blood from a neurosurgical (brain) operative field was used to identify peripherally-derived macrophages that showed strong Iba1+ but lacked TMEM119+ immunostaining (**P**–**R**). Scale bars are 25 μm. Unpublished data.

**Figure 6 biomolecules-12-00603-f006:**
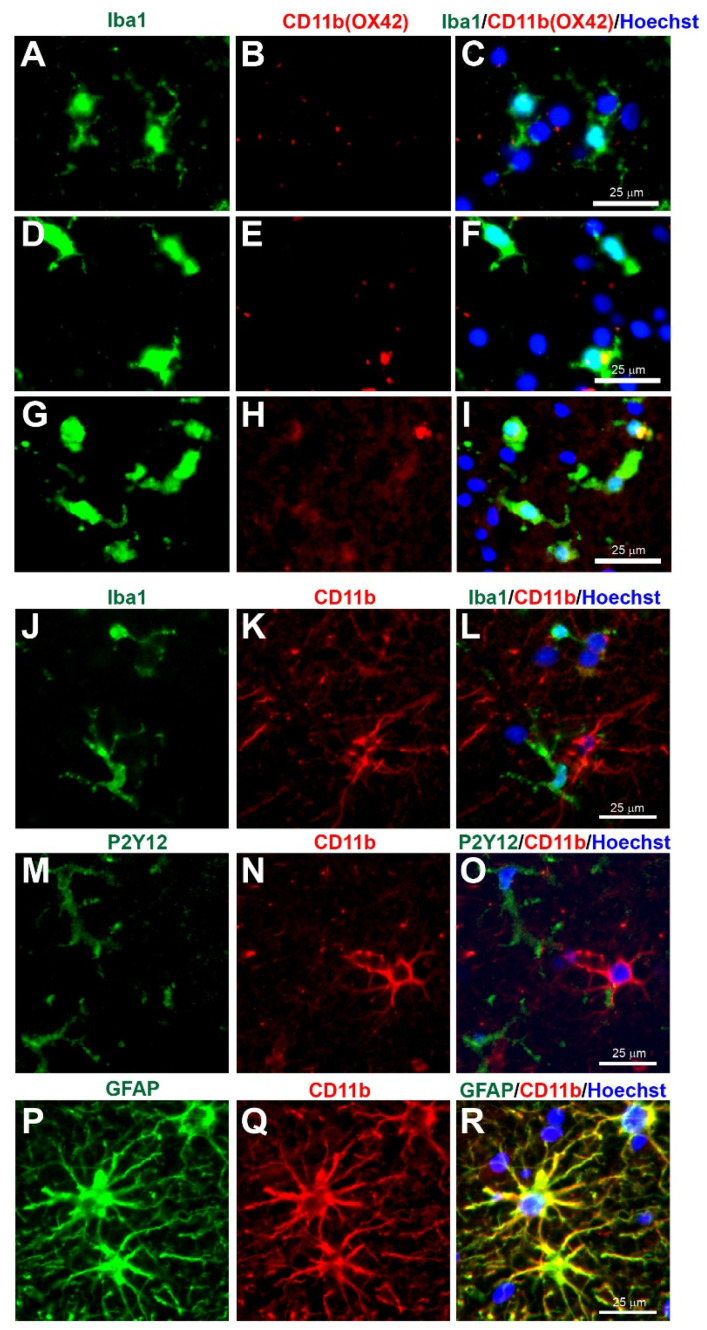
Concerns with CD11b (Clone OX42) immunostaining for microglia in human tissue. Human brain biopsy samples from the superior frontal gyrus after TBI were stained with the primary antibodies: mouse anti-rat OX42 (1:100, Bio_Rad, Cat. No. MCA275R) (**A**–**I**), or mouse anti-human CD11b (1:400, Novus Biologicals, Cat. No. MAB16991) (**J**–**R**); then with rabbit anti-Iba1 (1:1000, Wako, Cat No. 019-19741) (**J**), or rabbit anti-P2Y12 (1:200, AnaSpec, ANA55043A) (**M**), or rabbit anti-GFAP (1:1000, Dako, Cat No. Z0334) (**P**), followed by rabbit anti-Alexa Fluor 488 and mouse anti-Alexa Fluor 594 or tyramide amplification with ExtrAvidin FITC for the mouse anti-human CD11b (**K**,**N**,**Q**). Limited anti-rat OX42 expression was observed in Iba1+ microglia with a ramified state morphology (**A**–**C**), but some anti-rat OX42 expression in Iba1+ microglia in the more activated state (**D**–**F**) or contused brain tissue (**G**–**I**). Anti-human CD11b staining was absent in Iba1 (**J**–**L**) and P2Y12 (**M**–**O**) immunopositive cells, but coexpressed with the GFAP astrocytic marker (**P**–**R**). Scale bars are 25 μm. Unpublished data.

## Data Availability

Not applicable.

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
