# Peer review of "Advances in Visualizing Microglial Cells in Human Central Nervous System Tissue"

_biomolecules, 2022, doi:10.3390/biom12050603_

Round 1

Reviewer 1 Report

In the manuscript entitled "Advances in visualisation of microglia in tissue of the central nervous system", the authors introduce the topic of different microglial activity and activation states in various CNS conditions. They then describe a history of microglial staining techniques, beginning with silver stains from more than a century ago, proceeding to immunostains with markers for various macrophage-type cells (Iba1, CD68, HLA-DR, and Galectin-3), and culminating in the recently defined microglia-specific markers P2Y12 and TMEM119. Overall, I found the narrative to be informative and useful, although some of the details seemed to be lacking or incorrect. The major things that stuck out to me as needing improvement were 1) the inadequacy of Figure 1 and its legend, 2) the frequent references to M1 and M2 microglia, and 3) the absence of any description for the recently described TREM2-dependent DAM polarization state observed in many neurodegenerative disease models. In addition, the authors' treatment of TSPO as a microglial/inflammatory biomarker seemed either inadequate or out of place.

Major comments:

  1. Figure 1 shows some immunostains for Iba1 and P2Y12, but hardly any description is provided. The reader is supposed to just take it for granted that these represent different degrees of microglial morphological change ranging from fully ramified/resting to fully amoeboid/reactive, with the various states of activation being a result of acute traumatic brain injury. However, microglia in different brain regions (e.g., cortical GM versus cortical WM) can have very different morphologies even under normal conditions. Each picture might be from...anywhere in the brain? From an atrophic or degenerated area? From a penumbral area? From an unaffected healthy area? No details are provided about the location, timing, or duration of the TBI or its relation to the stained sections; the post-mortem interval and tissue acquisition; or tissue processing, staining, and imaging. Moreover, although P2Y12 is well-known for its high expression in resting microglia, it is drastically downregulated in virtually any microglial polarization state other than resting, so using P2Y12 to show the highly activated microglia in the fourth and fifth panels of the bottom row seems either inappropriate or highly questionable. What would be really fantastic would be to make a figure (or figures) showing several markers that the authors chose to highlight, co-stained with a stably expressed marker like Iba1, so that the reader can see what the immunolabeling for each marker actually looks like (CD68, HLA-DR, Galectin-3, P2Y12, TMEM119) in resting vs. activated cells. For example, this approach would show for CD68 that the immunostaining is barely detectable in normal microglia, whereas activated microglia in an injured brain region will show robust CD68 signal localized to the lysosome. Similarly, Gal-3 and HLA-DR will likely show elevated signal in activated microglia, while P2Y12 should do the opposite. TMEM119 will likely be present in both resting and activated microglia, underscoring its utility in labeling diverse states of microglial polarization (like Iba1, but even better since it is expressed by parenchymal microglia but not resident or infiltrating macrophages (Bennett 2016).) Another issue is that the sizes and the abundance of the DAPI-stained nuclei look pretty different among figure panels (compare top right vs. bottom right for size, or top left vs. top right for abundance), despite the scale bars all being identical length, so this makes it seem apparent that different regions are being imaged, with different numbers and subtypes of neurons being in the field, and these differences can definitely affect microglial morphologies. 
  2. The manuscript frequently refers to M1 and M2 microglia. This system of classification is badly outdated (Ransohoff, Nat Neurosci 19:987-991), especially since the purported M2-like microglial state (i.e., resembling the effect of IL4 or IL13 treatment) has not been observed in any in vivo condition or disease model that I am aware of (besides direct injection or overexpression of these cytokines). Many researchers have used a random marker or two to suggest that microglia are M2-polarized, but whenever I have seen this type of data, it always seems like poor quality immunostains or improperly interpreted PCR results, and I have not yet seen a convincing gene expression profile from purified brain microglia that resembles the expression profile of IL4/IL13-treated microglia. Arginase-1 and Mrc1 are popular examples of targeted analysis to determine M2 activation state, but most of these data are garbage. Arg-1 is barely expressed in microglia, and a 1.5-fold change in its microglial expression level is pretty meaningless compared to its expression level in monocytes being 50-fold or 100-fold higher than in microglia. As for Mrc1, it is a specific marker for PVMs in the brain, and microglia are Mrc1-negative. So when people try to measure changes in Mrc1 expression by PCR, they are actually just detecting PVM expression levels, not microglial expression. Referring to the M1 state is less problematic since microglia from LPS-treated mice do show many of the same changes that occur when microglia are treated with LPS in vitro. While the M1/M2 nomenclature persists in many corners, it doesn't make it right, and in my opinion the readers would be better served if the authors abandoned references to M2 microglia and made an effort to incorporate a modern view of microglial polarization states (e.g., see IR Holtman 2015 Acta Neuropath Comm 3:31, or BA Friedman 2018 Cell Reports 22:832-847). 
  3. The so-called DAM polarization state is perhaps the most commonly observed state of microglial activation, being present in virtually all models of neural damage and degeneration (see Deczkowska et al review in Cell 2018, and see the Friedman article mentioned above). Until this state was characterized in recent years, researchers often attempted to pigeonhole the microglia in degeneration models into either M1 or M2 classifications, both of which are terrible descriptors for the flavor of activation that occurs in these models. Some markers that are highly induced in DAM cells (and not induced in the classic M1/pro-inflammatory state) include Clec7A/Dectin-1, CD11c/Itgax, and LPL. It would be instructive for the authors to relate the DAM activation state to their chosen markers (upregulated CD68, Gal-3, MHC-II, and TSPO, and downregulated P2Y12).
  4. To me, the TSPO section felt a somewhat out of place since it is almost always used as a target for PET imaging while the rest of the manuscript seems all about staining microglia in fixed sections. I rarely if ever see TSPO used for immunostaining. Given the various challenges of TSPO PET ligands, it seems like these issues ought to be more fully described in order to do the subject justice, or at least refer the reader to one or two helpful, recent reviews that cover the subject fully. Some of these challenges include the lack of cell type specificity for TSPO expression, TSPO polymorphisms that affect ligand binding, uncertainty over whether increased TSPO signal indicates increased microglial cell number or increased TSPO expression, and TSPO being upregulated in both M1-like (damaging) and DAM (protective) microglial activation states. 

Minor comments: 

  • Line 39: Though it only recently (~12 years ago) became widely appreciated that microglia have an embryonic origin rather than being bone marrow-derived, there was pretty clear evidence for this published 20-30 years ago; however, not many people paid attention. Some reviews on the topic would highlight the earlier work.
  • Line 47:  "Perivascular microglia"-- This phrase is not really a thing. There are parenchymal microglia, and there are perivascular macrophages (PVMs). Both are CNS-resident cells derived from embryonic precursors. The microglia outnumber the PVMs by a factor of 25:1 or 50:1.
  • Line 49: The authors use the word 'implemented' when I think they meant 'implicated'.
  • Line 56:  'brian injury' instead of brain injury
  • Line 80:  "To fully appreciate the process in the visualisation of microglia..."  This is awkward phrasing that should be reworded.
  • Line 93:  States that Hortega's method was developed in 1932. This doesn't seem right to me since I recently read a review about Alzheimer's pathology, written in 1929, that referred to Hortega cells (an early name for microglia). So Hortega's method must have been developed at least by 1929.
  • Line 115: Azhiemr’s misspelled
  • Lines 120, 131:  "...the next biggest breakthrough..."  The phrase "next biggest" normally refers to the second largest in magnitude, not to a chronological sequence. Minimally, instances of this phrase should be changed to "the next big breakthrough...".  But rather than saying "the next big breakthrough" on multiple occasions, some other phrasing should be used.
  • Line 132:  "Iba1 is in a major histocompatibility complex (MHC) class III region of an EF-hand protein."  This sentence does not make sense. An MHC region refers to a region of a chromosome containing many, many genes. 
  • Line 150:  The word 'recently' should not be used in this sentence since the referenced stain by McGeer for MHC-II was over 30 years ago.
  • Lines 144 and 157 seem incongruent, with the former saying that even amoeboid microglia still have some processes and can be distinguished morphologically from infiltrating macrophages, while the latter says it is difficult to distinguish the two. Which is it?
  • Line 166:  "Interestingly, CD68 could be useful for studying microglia morphology due to intracellular expression." I disagree with this statement. CD68 protein is barely detectable in resting microglia, and in phagocytic microglia its strong detection is localized to the lysosome, so it is not at all useful for studying cellular morphology. In addition, it would be worth mentioning that under normal conditions, CD68 staining is easily visualized in PVMs but not in parenchymal microglia.
  • Lines 192-3:  TSPO increases a lot in M1-like microglia, and it increases modestly in DAM microglia (see Focke 2019 article in J. Nucl. Med.). Its lack of increase in M2 microglia is fairly meaningless since I don't believe M2 microglia are a real occurrence.
  • There are many instances of adjectival phrases that need to be hyphenated. For example, the phrase "microglia-specific" should always be hyphenated, as should phrases like "Iba1-positive".
  • Line 212:  "This raises the concern of whether all activation stages of microglia can be detected by P2Y12. If not, this could be a major limitation of this microglia marker."  This is not a question of 'if'. It is very clear that P2Y12 staining is lost in both M1-like and DAM microglia, so it is mainly the resting state that is effectively stained with this marker. 
  • TMEM119 section:  This section would benefit from a bit more description of the Bennett paper that originally characterized this marker. That paper showed that the marker persists in different microglial polarization states, so it may be generally useful like Iba1 but even better since it is microglia-specific. It would also be good for the authors to describe how TMEM119-GFP mice and TMEM119-Cre mice have been used for microglia-specific labeling and conditional knockout (PVMs are not labeled in TMEM119-GFP mice), and how these mice are more convenient and reliable than previously used LysM, CD11b, or Cx3cr1-driven schemes.
  • Along with TMEM119 being a marker that nicely labels microglia but not PVMs, it may also be worthwhile to mention markers that label PVMs but not microglia, including Mrc1/CD206 and LYVE1.

Author Response

In the manuscript entitled "Advances in visualisation of microglia in tissue of the central nervous system", the authors introduce the topic of different microglial activity and activation states in various CNS conditions. They then describe a history of microglial staining techniques, beginning with silver stains from more than a century ago, proceeding to immunostains with markers for various macrophage-type cells (Iba1, CD68, HLA-DR, and Galectin-3), and culminating in the recently defined microglia-specific markers P2Y12 and TMEM119. Overall, I found the narrative to be informative and useful, although some of the details seemed to be lacking or incorrect. The major things that stuck out to me as needing improvement were 1) the inadequacy of Figure 1 and its legend, 2) the frequent references to M1 and M2 microglia, and 3) the absence of any description for the recently described TREM2-dependent DAM polarization state observed in many neurodegenerative disease models. In addition, the authors' treatment of TSPO as a microglial/inflammatory biomarker seemed either inadequate or out of place.

We would like to thank the reviewer for taking time out of their busy schedule to provide a thorough review of this manuscript.

Major comments:

1. Figure 1 shows some immunostains for Iba1 and P2Y12, but hardly any description is provided. The reader is supposed to just take it for granted that these represent different degrees of microglial morphological change ranging from fully ramified/resting to fully amoeboid/reactive, with the various states of activation being a result of acute traumatic brain injury. However, microglia in different brain regions (e.g., cortical GM versus cortical WM) can have very different morphologies even under normal conditions. Each picture might be from...anywhere in the brain? From an atrophic or degenerated area? From a penumbral area? From an unaffected healthy area? No details are provided about the location, timing, or duration of the TBI or its relation to the stained sections; the post-mortem interval and tissue acquisition; or tissue processing, staining, and imaging.

Apologies for not providing further details for the review. We have now included a section on page 9 and within the figure legends to highlight the origin of the human brain biopsy samples and how they were processed. Although it would help to be able to stain microglia from different brain regions, it will be unethical to obtain multiple brain samples from living patients that had a severe traumatic brain injury (TBI).

2. Moreover, although P2Y12 is well-known for its high expression in resting microglia, it is drastically downregulated in virtually any microglial polarization state other than resting, so using P2Y12 to show the highly activated microglia in the fourth and fifth panels of the bottom row seems either inappropriate or highly questionable. What would be really fantastic would be to make a figure (or figures) showing several markers that the authors chose to highlight, co-stained with a stably expressed marker like Iba1, so that the reader can see what the immunolabeling for each marker actually looks like (CD68, HLA-DR, Galectin-3, P2Y12, TMEM119) in resting vs. activated cells. For example, this approach would show for CD68 that the immunostaining is barely detectable in normal microglia, whereas activated microglia in an injured brain region will show robust CD68 signal localized to the lysosome. Similarly, Gal-3 and HLA-DR will likely show elevated signal in activated microglia, while P2Y12 should do the opposite. TMEM119 will likely be present in both resting and activated microglia, underscoring its utility in labeling diverse states of microglial polarization (like Iba1, but even better since it is expressed by parenchymal microglia but not resident or infiltrating macrophages (Bennett 2016).)

We agree with the reviewer, so we have carried out the additional laboratory work and generated 6 new figures to show the co-expression of Iba1 with either: HLA-DR, CD68, OX42, Galectin-3, P2Y12 and TMEM119. Each figure contained microglia in a ramified state and 4 other activation states based on the morphology within human brain tissue. Furthermore, to distinguish microglia and macrophage, Clotted blood from the brain injury was also stained simultaneously with the brain biopsy to indicate the quality and selectivity of the antibodies used.

3. Another issue is that the sizes and the abundance of the DAPI-stained nuclei look pretty different among figure panels (compare top right vs. bottom right for size, or top left vs. top right for abundance), despite the scale bars all being identical length, so this makes it seem apparent that different regions are being imaged, with different numbers and subtypes of neurons being in the field, and these differences can definitely affect microglial morphologies. 

We agree with the reviewer, so we have removed the original figure 1 and replaced it with 6 more figures with correct magnifications used. As the brain biopsies at 10 mm x 1.5 mm in size were from the superior frontal gyrus, we are restricted to this brain region.

4. The manuscript frequently refers to M1 and M2 microglia. This system of classification is badly outdated (Ransohoff, Nat Neurosci 19:987-991), especially since the purported M2-like microglial state (i.e., resembling the effect of IL4 or IL13 treatment) has not been observed in any in vivo condition or disease model that I am aware of (besides direct injection or overexpression of these cytokines). Many researchers have used a random marker or two to suggest that microglia are M2-polarized, but whenever I have seen this type of data, it always seems like poor quality immunostains or improperly interpreted PCR results, and I have not yet seen a convincing gene expression profile from purified brain microglia that resembles the expression profile of IL4/IL13-treated microglia. Arginase-1 and Mrc1 are popular examples of targeted analysis to determine M2 activation state, but most of these data are garbage. Arg-1 is barely expressed in microglia, and a 1.5-fold change in its microglial expression level is pretty meaningless compared to its expression level in monocytes being 50-fold or 100-fold higher than in microglia. As for Mrc1, it is a specific marker for PVMs in the brain, and microglia are Mrc1-negative. So when people try to measure changes in Mrc1 expression by PCR, they are actually just detecting PVM expression levels, not microglial expression. Referring to the M1 state is less problematic since microglia from LPS-treated mice do show many of the same changes that occur when microglia are treated with LPS in vitro. While the M1/M2 nomenclature persists in many corners, it doesn't make it right, and in my opinion the readers would be better served if the authors abandoned references to M2 microglia and made an effort to incorporate a modern view of microglial polarization states (e.g., see IR Holtman 2015 Acta Neuropath Comm 3:31, or BA Friedman 2018 Cell Reports 22:832-847). 

We agree with the reviewer, so we have removed all the detailed discussions of M1 and M2 phenotypes and have included the references the reviewer has suggested.

5. The so-called DAM polarization state is perhaps the most commonly observed state of microglial activation, being present in virtually all models of neural damage and degeneration (see Deczkowska et al review in Cell 2018, and see the Friedman article mentioned above). Until this state was characterized in recent years, researchers often attempted to pigeonhole the microglia in degeneration models into either M1 or M2 classifications, both of which are terrible descriptors for the flavor of activation that occurs in these models. Some markers that are highly induced in DAM cells (and not induced in the classic M1/pro-inflammatory state) include Clec7A/Dectin-1, CD11c/Itgax, and LPL. It would be instructive for the authors to relate the DAM activation state to their chosen markers (upregulated CD68, Gal-3, MHC-II, and TSPO, and downregulated P2Y12).

We agree with the reviewer, so we added a section on disease-associated microglia on page 14.

6. To me, the TSPO section felt a somewhat out of place since it is almost always used as a target for PET imaging while the rest of the manuscript seems all about staining microglia in fixed sections. I rarely if ever see TSPO used for immunostaining. Given the various challenges of TSPO PET ligands, it seems like these issues ought to be more fully described in order to do the subject justice, or at least refer the reader to one or two helpful, recent reviews that cover the subject fully. Some of these challenges include the lack of cell type specificity for TSPO expression, TSPO polymorphisms that affect ligand binding, uncertainty over whether increased TSPO signal indicates increased microglial cell number or increased TSPO expression, and TSPO being upregulated in both M1-like (damaging) and DAM (protective) microglial activation states. 

We agree with the reviewer so we have removed the TSPO paragraph. Instead, we have added a section on OX42.

Minor comments: 

  • Line 39: Though it only recently (~12 years ago) became widely appreciated that microglia have an embryonic origin rather than being bone marrow-derived, there was pretty clear evidence for this published 20-30 years ago; however, not many people paid attention. Some reviews on the topic would highlight the earlier work.

Apologies. We have included the Alliot et al., (1999) paper to demonstrate their contribution to the origin of microglia from the yolk sac on page 2.

  • Line 47:  "Perivascular microglia"-- This phrase is not really a thing. There are parenchymal microglia, and there are perivascular macrophages (PVMs). Both are CNS-resident cells derived from embryonic precursors. The microglia outnumber the PVMs by a factor of 25:1 or 50:1.

Apologies. We have re-written the sentence on page 1.

  • Line 49: The authors use the word 'implemented' when I think they meant 'implicated'.

Apologies. We have corrected it as suggested.

  • Line 56:  'brian injury' instead of brain injury

Apologies. We have corrected the spelling.

  • Line 80:  "To fully appreciate the process in the visualisation of microglia..."  This is awkward phrasing that should be reworded.

Apologies. We have reworded this sentence on page 9.

  • Line 93:  States that Hortega's method was developed in 1932. This doesn't seem right to me since I recently read a review about Alzheimer's pathology, written in 1929, that referred to Hortega cells (an early name for microglia). So Hortega's method must have been developed at least by 1929.

Apologies. We have carried more detailed investigation and identified there was a series of papers by Hortega during 1919. It was a textbook by Hortega that was published in 1932.

  • Line 115: Azhiemr’s misspelled

Apologies. We have corrected the spelling on page 10.

  • Lines 120, 131:  "...the next biggest breakthrough..."  The phrase "next biggest" normally refers to the second largest in magnitude, not to a chronological sequence. Minimally, instances of this phrase should be changed to "the next big breakthrough...".  But rather than saying "the next big breakthrough" on multiple occasions, some other phrasing should be used.

Apologies. We have reworded this sentence on page 11.

  • Line 132:  "Iba1 is in a major histocompatibility complex (MHC) class III region of an EF-hand protein."  This sentence does not make sense. An MHC region refers to a region of a chromosome containing many, many genes. 

Apologies. We have reworded this sentence on page 11.

  • Line 150:  The word 'recently' should not be used in this sentence since the referenced stain by McGeer for MHC-II was over 30 years ago.

Apologies. We have re-written this sentence on page 11.

  • Lines 144 and 157 seem incongruent, with the former saying that even amoeboid microglia still have some processes and can be distinguished morphologically from infiltrating macrophages, while the latter says it is difficult to distinguish the two. Which is it?

Apologies. We have re-written this sentence on page 11.

  • Line 166:  "Interestingly, CD68 could be useful for studying microglia morphology due to intracellular expression." I disagree with this statement. CD68 protein is barely detectable in resting microglia, and in phagocytic microglia its strong detection is localized to the lysosome, so it is not at all useful for studying cellular morphology. In addition, it would be worth mentioning that under normal conditions, CD68 staining is easily visualized in PVMs but not in parenchymal microglia.

We may have to partially agree with the reviewer on this point as based on our data (figure 2) and Hendrickx et al., 2017 we do show expression of CD68 with ramified microglia. Therefore, we included a statement to indicate this difference in opinions on page 12.

  • Lines 192-3:  TSPO increases a lot in M1-like microglia, and it increases modestly in DAM microglia (see Focke 2019 article in J. Nucl. Med.). Its lack of increase in M2 microglia is fairly meaningless since I don't believe M2 microglia are a real occurrence.

Apologies. We have removed the TSPO section suggested above.

  • There are many instances of adjectival phrases that need to be hyphenated. For example, the phrase "microglia-specific" should always be hyphenated, as should phrases like "Iba1-positive".

Apologies. We believe we have hyphenated all the relevant phrases.

  •  
  • Line 212:  "This raises the concern of whether all activation stages of microglia can be detected by P2Y12. If not, this could be a major limitation of this microglia marker."  This is not a question of 'if'. It is very clear that P2Y12 staining is lost in both M1-like and DAM microglia, so it is mainly the resting state that is effectively stained with this marker. 

Apologies. We have re-written this section on page 13 as we have supporting evidence to suggest P2Y12 in our human brain injury samples that P2Y12 can still effectively label microglia irrelevant to the activation state.

  • TMEM119 section:  This section would benefit from a bit more description of the Bennett paper that originally characterized this marker. That paper showed that the marker persists in different microglial polarization states, so it may be generally useful like Iba1 but even better since it is microglia-specific. It would also be good for the authors to describe how TMEM119-GFP mice and TMEM119-Cre mice have been used for microglia-specific labeling and conditional knockout (PVMs are not labeled in TMEM119-GFP mice), and how these mice are more convenient and reliable than previously used LysM, CD11b, or driven .

We agree with the review, so we have provided further information regarding the study by Bennett and colleagues on page 13.

  • Along with TMEM119 being a marker that nicely labels microglia but not PVMs, it may also be worthwhile to mention markers that label PVMs but not microglia, including Mrc1/CD206 and LYVE1.

We think that including a section on PVM markers would distract the readers that do not have expertise with microglia but are interested in entering the neuroinflammation field.

Reviewer 2 Report

In the present paper by Uff et al. with the title “Advances in visualisation of microglia in tissue of the central nervous system”, the reader is presented with a comprehensive review of microglia markers for visualising microglia in CNS tissue samples and will be appreciated by a broad readership. Indeed, there is a growing body of literature that recognises the importance of microglia and its functional roles within the CNS during homeostasis and pathophysiological conditions, therefore the identification of microglia phenotype under certain condition is crucial. Generally, I support the publication of the review paper. Nevertheless, there are a few minor comments that would require attention and I encourage the authors to address them before the paper is accepted for publication.

Main comments:

  1. As authors stated, it is important to expose the awareness of the microglia-selectivity issues of the various stains and immunomarkers used by researchers to distinguish microglia in CNS tissue to avoid misinterpretation. Therefore, it would be beneficial to include in the introduction the properties (expression of certain proteins) that distinguish between ramified/resting microglia (M2 phenotype) and reactive form of microglia (M1 phenotype), on which the phenotyping of microglia is based.
  2. In relation to the above point, it would be sensible to also define microglial markers for each phenotype (M1, M2), maybe in additional table.
  3. Some well-used microglial markers are missing, such as OX-6, OX-42 (Cd11b, Cd11c) for M1 phenotype, CD206, arginase-1 for M2 phenotype.

Minor points:

  1. Line 55-57 – Figure's legend of Figure 1 should be listed below the panels; please also indicate which microglia cells were used.
  2. Line 81 - I am little confused as authors stated to provide the historical timeline of microglial staining from 1906 until 2021; however the authors describe microglia discovery by Franz Nissl and using his eponymous Nissl stain to identify cells in 1899.
  3. Table 1 – why the companies that produce certain markers are listed in the Table? Are they the only ones to provide such markers? Please, reconsider the structure of the Table 1.
  4. Line 115 – typo error: AD, Alzheimer's

Author Response

In the present paper by Uff et al. with the title “Advances in visualisation of microglia in tissue of the central nervous system”, the reader is presented with a comprehensive review of microglia markers for visualising microglia in CNS tissue samples and will be appreciated by a broad readership. Indeed, there is a growing body of literature that recognises the importance of microglia and its functional roles within the CNS during homeostasis and pathophysiological conditions, therefore the identification of microglia phenotype under certain condition is crucial. Generally, I support the publication of the review paper. Nevertheless, there are a few minor comments that would require attention and I encourage the authors to address them before the paper is accepted for publication.

 We would like to thank the reviewer for taking time out of their busy schedule to provide a thorough review of this manuscript.

Main comments:

1. As authors stated, it is important to expose the awareness of the microglia-selectivity issues of the various stains and immunomarkers used by researchers to distinguish microglia in CNS tissue to avoid misinterpretation. Therefore, it would be beneficial to include in the introduction the properties (expression of certain proteins) that distinguish between ramified/resting microglia (M2 phenotype) and reactive form of microglia (M1 phenotype), on which the phenotyping of microglia is based.

As strongly advised by reviewer 1 and also with our agreement, we have not included any discussion on the outdated M1 and M2 phenotype concept.

2. In relation to the above point, it would be sensible to also define microglial markers for each phenotype (M1, M2), maybe in additional table.

In reference to comment 1, we do not think it is advisable to discuss M1 and M2 phenotypes in detail.

3. Some well-used microglial markers are missing, such as OX-6, OX-42 (Cd11b, Cd11c) for M1 phenotype, CD206, arginase-1 for M2 phenotype.

 We have provided some new figures, including OX42 co-stained with Iba1 (figure 3). As for OX6, arginase-1 and CD206, it has been suggested not to be expressed by all microglia (PMID: 23313316), and/or not microglia specific (PMID: 25386179; PMID: 32002295) so the addition of these molecules in the discussion will further confuse readers that have limited knowledge of microglia.

Minor points:

1. Line 55-57 – Figure's legend of Figure 1 should be listed below the panels; please also indicate which microglia cells were used.

Apologies. We correct this error.

2. Line 81 - I am little confused as authors stated to provide the historical timeline of microglial staining from 1906 until 2021; however the authors describe microglia discovery by Franz Nissl and using his eponymous Nissl stain to identify cells in 1899.

Apologies. We altered the date to 1899.

3. Table 1 – why the companies that produce certain markers are listed in the Table? Are they the only ones to provide such markers? Please, reconsider the structure of the Table 1.

We have included the companies because from experience, not all the antibodies sold by companies are of the same quality. For example, many companies make antibodies against Iba1, but the Iba1 from Wako provides the best staining in comparison to the others that I have personally tested. Therefore, providing this information will benefit the readers that are interested in staining for microglia.

4. Line 115 – typo error: AD, Alzheimer's

Apologies. We correct this error.

Round 2

Reviewer 1 Report

The revised version of the review manuscript entitled "Advances in visualisation of microglia in human tissue of the central nervous system" by Uff, Patel, and Yip is greatly improved compared to the original. Overall, the authors have adequately addressed all of my concerns and recommendations for improvement. There is one thing that must be corrected before it is published, which is some erroneous immunostaining.

Most of the immunostains are a valuable improvement for the manuscript, but the OX42 stain is wrong. OX42 is the name of a mouse monoclonal antibody that only recognizes rat CD11b. To my knowledge, it does not recognize human CD11b at all. The name of the immunostaining target should be CD11b or ITGAM; it should never be called OX42. The immunostains that show some OX42 reactivity in human tissue are an example of the authors finding weak non-specific signal and attributing it to the intended target of OX42 reactivity. I recommend that the authors repeat the staining with a human-reactive CD11b antibody. The results should look more similar to the P2RY12 and TMEM119 stains, except that the non-microglial myeloid cells in the blood clot field, which are negative for the microglia-specific markers P2RY12 and TMEM119, should be positive for CD11b.

A minor recommendation on the title:  I think "Advances in visualizing microglial cells in human central nervous system tissues" will sound better. The current title makes it sound like 'human tissue' is a component of the central nervous system. And 'visualizing microglial cells' is active instead of passive phrasing. 

Finally, I would like to thank the authors for maintaining their original stance regarding my comment about "perivascular microglia". I had not seen a few pieces of important literature published in the last few years on this topic, and thanks to the authors' manuscript I endeavored to learn more about it and now I am better informed.
